# Measuring the Efficiency of Economic Growth towards Sustainable Growth with Grey System Theory

**Małgorzata Kokocińska [1], Marcin Nowak [2] and Paweł Łopatka [3,*]**

[1] Institute of Economics and Finance, Faculty of Economics and Management, University of Zielona Góra ul. Licealna 9, 65-417 Zielona Góra, Poland; m.kokocinska@wez.uz.zgora.pl

[2] Faculty of Engineering Management, Poznan University of Technology, pl. Marii Skłodowskiej-Curie 5, 60-965 Poznań, Poland; marcin.nowak@put.poznan.pl

[3] Institute of Economics, Department of Microeconomics, Poznań University of Economics and Business, al. Niepodległości 10, 61-875 Poznań, Poland

[*] Correspondence: pawellopatka.uep@gmail.com

**Abstract:** In the paper, a new indicator exemplifying the conversion efficiency of expenditures towards economic growth into results pertaining to sustainable development, dubbed the "Synthetic Efficiency Indicator for Economic Growth" (hereinafter: "SEI-EG") has been proposed. The inspiration for proposing such an indicator was the identification of the lack of connections between research on economic convergence and the research area connected with sustainable growth category. It was assumed that, in the first place, outcomes of the proposed convergence will be visible in developed economies, represented by EU15 member states. The set goal was to provide an answer to the question of difference between EU15 member states with respect to efficiency of converging expenditures exemplifying economic growth into results pertaining to sustainable growth. The research was conducted for 2016–2018 using Grey System Theory. With the use of the elaborated indicator, the authors created a ranking list of countries based on the efficiency of economic growth towards sustainable growth criterion. The conducted research proved that, in general, the smaller EU member states are characterized by significantly higher efficiency of converging expenditures exemplifying economic growth into results pertaining to sustainable development in the researched area. Among the countries with large economies, only Germany showed efficiency comparable to smaller ones.

**Keywords:** economic convergence; sustainable development; Grey Whitenization Model

## 1. Introduction

Economic convergence is one of the most important issues in comparative research of countries. Most often, these studies are focused on the process of catching-up of the less developed countries with the more developed ones (beta convergence), as well as regional differentiation within individual countries (gamma convergence). Research on economic convergence has evolved from analyses of entire economies to regional analyses, especially with the emergence of the European Union's regional policy. It has become an important element in assessing differences between countries and regions, not only in terms of the rate of GDP growth itself but also in terms of growth factors [1,2]. These analyses are nowadays continued in studies on the links between economic growth and sustainable development, such as causal linkages between the quality of country-level governance, economic growth and a well-known indicator of economic sustainable development [3–5].

The notion of economic convergence is strictly related to more general aspects of long-term development as well as sustainable growth. The evolution of theoretical concepts and empirical studies on convergence has resulted in a considerable broadening of the scope of research [6,7]. While earlier studies have shown that there is a distinctiveness of the classical approaches to convergence and of the comparative analysis in the area of sustainable growth, the emphasis is nowadays on the relationship between economic growth and sustainable development [8]. As emphasized, assessing development policy solely on the basis of convergence criteria, as was done with the EU Cohesion Policy for example, makes little or no sense [9] since convergence does not capture the socioeconomic objective of the policy, which is to emphasize institutional and learning behavior [10].

Thus, the link between convergence and sustainable development can be described as sustainable convergence. What is beyond dispute is the change in the perception of economic development objectives and the shift from the concept of economic growth itself to sustainable development. The very definition of sustainable development and the definition of appropriate indicators are crucial. What is considered to be crucial is the report [11] and the formulation: "Humanity has the ability to make development sustainable to ensure that it meets the needs of the present without compromising the ability of future generations to meet their own needs". The Europe 2000 project [12] points to an extended definition: "Sustainable growth means building a resource efficient, sustainable and competitive economy, exploiting Europe's leadership in the race to develop new processes and technologies, including green technologies, accelerating the roll out of smart grids using ICTs, exploiting EU-scale networks, and reinforcing the competitive advantages of our businesses, particularly in manufacturing and within our SMEs, as well through assisting consumers to value resource efficiency. Such an approach will help the EU to prosper in a low-carbon, resource constrained world while preventing environmental degradation, biodiversity loss and unsustainable use of resources. It will also underpin economic, social and territorial cohesion."

Moreover, in up-to-date studies one is more likely to notice the distinctiveness of considerations on classical approaches to convergence and the distinctiveness of comparative analyses in the sustainable growth field. The second approach is favored by numerous and complex classifications of sustainable growth indicators [13], as well as the introduction in publications of the "social convergence" notion, within which the analyses cover aspects of economy like living standards, regional inequality, well-being [14,15].

Adopted in September 2015, The Agenda for Sustainable Development 2030 is a comprehensive development plan for the world with a 2030 perspective established by the UN, through negotiations between the Member States. The adoption of the Agenda 2030 is an event unprecedented in human history. All 193 UN member countries have committed themselves to the 17 Sustainable Development Goals (SDGs). Most of these goals can be reduced to a set of indicators, relevant to developed economies [16] that were used in the study.

The authors' proposed and integrated approach to convergence research is to elaborate an indicator to measure outcomes of economic growth with benefits towards sustainable growth. It has been adopted that conversion of the expenditure category, represented by the GDP (per capita in Purchasing Power Standards, hereinafter: PPS per capita) and the general government gross debt (percentage of GDP) into results connected with sustainable development indicators in three key areas, i.e., Industry, Innovation and Infrastructure, should constitute an answer to the question of whether the countries, in the long-term, achieving relatively higher economy growth stage are characterized by higher efficiency of economy transformation towards sustainable growth. Due to the said multiplicity and considerable de-aggregation of sustainable development indicators, only nine symptomatic indicators from the Eurostat base were selected, representing the three areas indicated above: industry, innovation and infrastructure [17]. In particular, these are the following indicators: gross domestic expenditure on R&D, employment in high- and medium-high technology manufacturing and knowledge-intensive services, R&D personnel, patent applications to the European Patent Office, share of buses and trains in total passenger transport, share of rail and inland waterways in total freight transport, and average $CO_2$ emissions per km from new passenger cars [17]. The selection of these indicators—besides a general consensus as to their importance for modern

economies—is also justified from the point of view of economic productivity and efficiency of public investments while, in the long-term, interest rates remain low. The dramatically low interest rates and low capital costs are good conditions for modernization of infrastructure, investments in education and support for green technologies. There is a supposition that social rate of profit from those investments is many times higher than the current public debt cost [18]. Except for the selection of indicators, from the point of view of research procedure, the important aspect is also the selection of countries for analysis. It has been adopted, that the so-called convergence clubs, that is informal groups of countries evidencing considerable similarities, are of importance here. Taking into account the job market (working people who generate national product) and the level of GDP per capita in PPS in the available 2018 data, one may notice that there are two distinct groups of large and smaller EU15 member states [4]. A constantly held and long-established view is that smaller countries are in a worse position, for they do not feature a differentiated economic structure, strong army, bargaining power in negotiations, and are less resistant to crises. Smaller European Union member states also face size-related challenges in the EU multilevel system, such as in day-to-day policymaking [19,20]. However, in publications on the subject, there is the change in function of time visible, consisting in moving the center of mass in economic growth evaluations of smaller countries towards the notion of a small but smart country [21], then towards connection of smaller country's functioning with entrepreneurship [22], and finally with the answer to the question of how "smallness" may be defined more positively. Currently, "smallness" is replaced with the marker of a smart and innovative country, and the politicians claim that following the Cold War the differences between large and smaller countries are more and more becoming less visible [23,24]. Authors of this elaboration make an assumption that in the period without wars, smaller countries evidence institutional and economic advantage, are characterized by faster convergence in time and earlier transition towards sustainable growth stage. The potential advantage factor of smaller countries over large ones may be the very efficiency of gradual conversion of expenditures exemplifying GDP (per capita in PPS) and the general government gross debt (percentage of GDP) into results connected with sustainable development indicators.

By adopting such an assumption, three main research objectives have been formulated:

- Elaboration of "Synthetic Efficiency Indicator for Economic Growth" (SEI-EG) determination methodology. To complete this objective, the project method has been used.
- Determination of the synthetic efficiency indicator for economic growth for 11 EU member states belonging to two convergence groups. The research covers 2016–2018 macro-economic data for the analyzed countries. To complete the second objective, the Grey Whitenization Model has been used, being one of the basic theoretical constructs in Grey System Theory.
- Explanation of whether smaller countries are better in converting the economic growth results into results in sustainable growth area, based on the determined synthetic efficiency for economic growth ranking list for the analyzed countries.

The research problem undertaken in the current project is of a socio-economic nature. The data describing such systems are often small in size and burdened with incompleteness and subjectivity. In the empirical part of the article, data in the form of short time series related to the issue of sustainable development are analyzed. In principle, therefore, they are small in size. Statistical analysis often becomes an inadequate tool for modeling uncertainty in such cases. In the article, in order to solve the research problem, the methods included in the theory of grey systems, which are indicated in the literature on the subject as appropriate for the analysis of socio-economic systems [25] were used. This made it possible to draw conclusions on the basis of samples which, on the one hand, are small (the analysis included 15 countries), and on the other hand, are described by short time series (data for the period 2016–2018 was included). This approach to the research problem contributed both to the theoretical contribution to the development of new indicators describing sustainable development, and to the contribution of empirical research results linking sustainable development to the issue of economic convergence.

The paper has been organized in the following way. In the first part, the criteria connected with the occurrence of convergence groups within the EU are presented, together with the evolution in

perception of economic potential of smaller countries, and bases for Grey System Theory are discussed. In the second part, the methodology of SEI-EG determination is presented. In the third part, the results of the conducted research are presented, followed by the conclusions and directions for further research.

## 2. Materials and Methods

Methods included in Grey System Theory (hereinafter: "GST") are a tool for modeling information uncertainty, complementary to the probability theory, fuzzy logic or rough set theory [26,27]. GST is a comparatively young theory—the first paper in English on grey systems was published in 1989 by J. Deng [28]. Methods included in the GST are widely used in practice, especially when information uncertainty manifests in the form of a small set of data (which renders it impossible to use statistical methods—e.g., with respect to short time periods), incomplete information or information burdened with inaccuracy or subjectivity [29]. The described application area of methods included in Grey System Theory spurred their growing popularity [30]. In recent years, using a number of grey models, the uncertainty within grey systems with economic, social, technical or even natural character has been modeled [31,32].

In the paper [25], the authors focused on the problem of adequacy of the application of methods included in the theory of grey systems in modeling uncertainty in socio-economics systems. The main conclusion from the analyses was that the possibility of using data that are few, incomplete and subjective, makes GST an adequate tool for modeling systems whose key element is the human being. Moreover, most of the benefits of GST are achieved by modeling socio-economic systems. At the same time, the authors present a broad bibliometric analysis of the use of GST in analysis, assessment, prediction and relationship research in socio-economics systems.

In recent years, GST has been applied, among others, to: human resources management [33,34] healthcare management [35], customer satisfaction assessment [36] and labor migration management [37]. A particularly explored area of application of GST is sustainable development issues. The definition of policies on many aspects of sustainable development using the GST methodology has been the subject of many projects such as: [32,38–41].

Among grey system models there are, e.g., grey prediction models, grey decision-making models, grey control models, and grey relations models [37]. The selected methods of information uncertainty modeling are included in Grey System Theory, when one of the three constructs is used in them—grey numbers, whitenization function, or distributed grey [30].

The grey number is a number for which exact value is not known, but the interval of the number is known. The grey number is presented the following way:

1) $\otimes \rightarrow \left[\,\underline{a}, \overline{b}\,\right]$—for interval number
2) $\otimes \rightarrow \{a_1, \ldots, a_i, \ldots a_n\}$—for discrete number

The $\underline{a}$ symbol means lower interval limit, in which the given interval number is found, and the $\overline{b}$ symbol means upper interval limit, in which the given interval number is found. Although the following dependence is required $\underline{a} \leq \overline{b}$. The $a_1$ symbol means the first element of the set, in which the given discrete number is found, and the $a_i$ symbol means ith element of the set. Although the dependence is required $a_{k-1} \leq a_k \leq a_{k+1}$.

Definition 1. The grey interval number $\otimes G$ n is a real number d*, which meets the following condition: $\left\{d^* \in \left[\,\underline{a}, \overline{b}\,\right]\right\} \wedge \left\{\underline{a} \neq \overline{b}\right\} \wedge \left\{(\underline{a} \vee \overline{b}) \neq {}^{\pm}\infty\right\}$.

The issue of efficiency indicator value determination for the analyzed object (or a set of objects) is the issue of generating white number (exactly one value) from grey number (one value that is unknown, but its interval is known). In Grey System Theory, this process is often reduced to whitenization function determination and its calculation on the basis of the as-is empirical base. The whitenization function may be defined in the following way: if we assume we have a given grey space $\{\Omega, F, G^\circ\}$ and set $A \in F$, and set $Y = \{y \in R : 0 \leq y \leq 1\}$, then the function f: $A \rightarrow Y$ is dubbed the whitenization function, when it meets the following conditions [42]:

1) f(A) = 1, for every A= {a$_i$}, that is for every single-element set
2) f(∅) = 0

The whitenization function is a part of a wider construct, that is the grey space. In the grey space {Ω, F, G°}:

- Ω means a set being the analysis space.
- F will be σ—an algebra generated on set Ω, i.e.,: ∅ ∈ F and A ∈ F ⇒ A′ ∈ F and A$_1$, A$_2$, A$_3$ ∈ F ⇒ $\bigcup_{i=1}^{\infty}$ A$_i$ ∈ F.
- G° will be the function mapping F into a set of real numbers, which may be formally noted as G°: F→R, and will be the grey function, when it meets four axioms.

Axiom 1. G° (A$_i$) ≥ 0 for every A$_i$ ∈ F

Axiom 2. G° (A$_i$) = 0 ⇔ A$_i$ = {a$_i$}, with A$_i$ = {a$_i$} being any given single-element set belonging to F

Axiom 3. G° (Ω) = 1

Axiom 4. G° ($\bigcup_{i \in N}$ A$_i$) = $\sum_{i \in N}$ G°(A), where: A$_i$ ∩ A$_j$ = ∅ ∧ i ≠ j ∧ A$_i$ ≠ { a$_i$ }

From the structural point of view, the synthetic efficiency indicator for economic growth is a special instance of an efficiency indicator. In the theory of economy, efficiency is most often defined as a ratio of the obtained results to expenditures in a set of analyzed objects [43]. These objects may be both countries and companies, or production plants. For the purpose of this elaboration, an assumption has been adopted that efficiency is a relative measure, determined on the basis of the as-is empirical base. In addition, normalization of the efficiency indicator in interval [0,1] has been assumed. As assumed, each object being a subject of the research features some concrete value of the efficiency indicator. This value is not known, but the interval of the value is known (interval [0,1]). Thus, the cognitive uncertainty pertaining to subject's efficiency has been presented as a grey number. Grey numbers are the subject of Grey System Theory, one of the uncertainty modeling methods that is gaining increasing popularity. The elaborated methodology is connected with determination of the whitenization function, which makes it possible to determine one concrete value of the synthetic indicator for sustainable economic growth. The elaborated grey model of whitenization may be presented as a procedure consisting in 7 steps.

## 2.1. Step 1. Determination of Basic Elements of Whitenization Model

In the first step, the subject of whitenization must be defined. In the proposed model, the subject of whitenization process will be the synthetic efficiency indicator for economic growth. For each of the subjects, its value is a grey number ⊗ ∈ [0; 1]. As a result of the whitenization process, each of the analyzed subjects is assigned with efficiency indicator, being a white number (a number as understood traditionally). In the second step of the proposed model, the set of analyzed objects must be defined. In the third step, sets of expenditures and results constituting the basis for determination of a synthetic indicator for sustainable economic growth in the analyzed set of objects must be determined. The result of the first step is the data matrix *D* with the following form:

$$
D = [d_{ik}] = \begin{bmatrix}
r_{11} & r_{12} & \cdots & r_{1m} \\
r_{21} & r_{22} & \cdots & r_{2m} \\
\vdots & \vdots & \ddots & \vdots \\
r_{n1} & r_{n2} & \cdots & r_{nm} \\
i_{11} & i_{12} & \cdots & i_{1m} \\
i_{21} & i_{22} & \cdots & i_{2m} \\
\vdots & \vdots & \ddots & \vdots \\
i_{j1} & i_{j2} & \cdots & i_{jm}
\end{bmatrix}
\tag{1}
$$

where: *D*—data matrix; $r_{ik}$—ith result for kth object, *i* = 1,2,…, n, *k* = 1,2,…, m; $i_{ik}$—ith expenditure for kth object, *i* = 1,2,…, j, *k* = 1,2,…, m

## 2.2. Step 2. Elaboration of Scaled Matrices of Input Data $D^*$

In the second step of the proposed method, the matrix of scaled input data is elaborated with the form of (2).

$$
D^* = [d_{ik}^*] =
\begin{bmatrix}
r_{11}^* & r_{12}^* & \cdots & r_{1m}^* \\
r_{21}^* & r_{22}^* & \cdots & r_{2m}^* \\
\vdots & \vdots & \ddots & \vdots \\
r_{n1}^* & r_{n2}^* & \cdots & r_{nm}^* \\
i_{11}^* & i_{12}^* & \cdots & i_{1m}^* \\
i_{21}^* & i_{22}^* & \cdots & i_{2m}^* \\
\vdots & \vdots & \ddots & \vdots \\
i_{j1}^* & i_{j2}^* & \cdots & i_{jm}^*
\end{bmatrix}
\tag{2}
$$

where $d_{ik}^*$ means:

1.  For data being maximands:

$$
d_{ik}^* = \frac{d_{ik}}{d_{ik}^{min}}
\tag{3}
$$

2.  For data being minimands:

$$
d_{ik}^* = \frac{d_{ik}^{min}}{d_{ik}}
\tag{4}
$$

## 2.3. Step 3. Elaboration of Vector of Synthetic Expenditure Indicators $I_k$ for all Decision-Making Objects

In the proposed method, it is assumed that for expenditures describing the given decision-making object, a synthetic expenditure indicator is generated. Synthetic expenditure indicator $I_k$ is determined with the Formula (5).

$$
I_k = \sum_{i=1}^{j} i_{ik}^*
\tag{5}
$$

where:

- $I_k$ — synthetic expenditure indicator for kth decision-making object.
- $i_{ik}^*$ — scaled ith expenditure of kth decision-making object.

The vector of synthetic expenditure indicators $I_k$ has the following form:

$$
[I_1, I_2, \ldots, I_m]
\tag{6}
$$

## 2.4. Step 4. Elaboration of E Partial Efficiency Matrices

In the second step of the method, a matrix of partial efficiencies of decision-making objects E is elaborated based on the Formula (7).

$$
E = [e_{ik}] =
\begin{bmatrix}
e_{11} & e_{12} & \cdots & e_{1m} \\
e_{21} & e_{22} & \cdots & e_{2m} \\
\vdots & \vdots & \ddots & \vdots \\
e_{n1} & e_{n2} & \cdots & e_{nm}
\end{bmatrix}
\tag{7}
$$

where:

- $E$ — matrix of partial efficiencies of the researched decision-making objects.
- $i = 1,2,\ldots,$ m — determination of partial efficiency indicator.
- $k = 1,2,\ldots,$ n — determination of decision-making objects.
- $e_{ik}$ — ith partial efficiency indicator for kth decision-making object.

Partial efficiency $e_{ik}$ is determined with the use of the Formula (8).

$$e_{ik} = \frac{r_{ik}^*}{I_k} \qquad (8)$$

where:

- $r_{ik}^*$—scaled value of ith result for kth object.
- $I_k$—value of synthetic expenditure indicator for kth decision-making object.

### 2.5. Step 5. Determination of Reference and Anti-Reference Vector of Partial Efficiency

In the elaborated method, it has been assumed that efficiency is a relativized measure towards the as-is empirical base. The empirical data are the basis for determination of reference vector and anti-reference vector.

The reference vector is determined through a set-up of the highest values of individual partial efficiency indicators, irrespective of which object obtained them. The vector may be represented with the Formula (9).

$$REF = \begin{bmatrix} e_{1max} \\ e_{2max} \\ \vdots \\ e_{nmax} \end{bmatrix} \qquad (9)$$

The object with partial efficiency vector equaling the reference vector, would feature full efficiency of 1.00.

The anti-reference vector is determined through a set-up of the lowest values of individual partial efficiency indicators, irrespective of which object obtained them. The vector may be represented with the Formula (10).

$$AREF = \begin{bmatrix} e_{1min} \\ e_{2min} \\ \vdots \\ e_{nmin} \end{bmatrix} \qquad (10)$$

The object with partial efficiency vector equaling the anti-reference vector, would feature efficiency of 0.00.

### 2.6. Step 6. Standardization of Partial Efficiency Matrices $E^*$ to Interval (0,1)

In the fifth step of the whitenization method, all values of the standardized partial efficiency matrix are normalized to interval (0,1), with the Formula (11).

$$e_{ik}^* = \frac{[e_{ik} - \min(e_{ik})] \cdot (e_{max}^* - e_{min}^*)}{\max(e_{ik}) - \min(e_{ik})} + e_{min}^* \qquad (11)$$

where:

- $\min(e_{ik})$ —minimum value of ith partial efficiency in the set of all objects.
- $\max(e_{ik})$—maximum value of ith partial efficiency in the set of all objects.
- $e_{max}^*$—the assumed maximum value of the standardized partial efficiency.
- $e_{min}^*$—the assumed minimum value of the standardized partial efficiency.

### 2.7. Step 7. Determination of Synthetic Indicator for Sustainable Economic Growth for Each Decision-Making Object

The efficiency of each decision-making object falls between the efficiency determined by the reference vector and anti-reference vector. In this model, it has been proposed to use a whitenization function, which may be presented in a graphical form on a radar diagram. In the radar diagram, the number of axes that equal the number of partial efficiencies is featured. The beginning of the axis for each of the standardized partial efficiencies from the matrix $E^*$ is determined by the minimum standardized partial efficiency contained in the anti-reference vector (AREF) set. The vertex of the

radar diagram is determined, on the other hand, by the maximum partial efficiency contained in the REF vector. In each axis, a relative value of partial efficiency of a given object towards the empirical base is plotted. In the case of the analyzed decision-making model, the weighted whitenization function for the object kth assumes the following form (12):

$$f(E^*) = \frac{S_k}{S_{REF}} \tag{12}$$

where:

- $f(E^*)$—whitenization function (assigns a grey number with a value of a white number).
- $S_k$—area of a polygon determined by values stemming from a vector describing standardized partial efficiencies of the kth object.
- $S_{REF}$—area of a polygon determined by values stemming from the standardized reference vector.

The value $S_k$ may be determined in a number of ways. One of them is to employ an analytical method of calculating the areas of polygons (Gaussian elimination):

$$F = \frac{1}{2} \left| \sum_{i=1}^{n} X_i(Y_{i+1} - Y_{i=1}) \right| \tag{13}$$

where:

- F—calculated area.
- $X_i, Y_i$—coordinates of the ith vertex; vertices are numbered one by one, from 1 to n.

As a result of using the weighted whitenization function, each of the decision-making objects is assigned with a whitened value of the efficiency indicator out of the interval (0,1).

## 3. Results

In research on economic convergence in the European Union, the dominant studies are analyses covering EU28, EU15, EU10 member states or their narrowed groups, for example V4 or large countries characterized by a considerable economy of scale [4,44]. In this paper, as discussed in the Introduction, the selection of countries for the purpose of the analysis was driven by identification of countries with division into smaller and large ones within the most developed group of EU15 member states. This group generally meets the condition of, on average, a higher level of GDP per capita in comparison to other EU member states. It is assumed that reaching a relatively high level of economic growth is the basic prerequisite for actions in favor of sustainable growth. Additionally, the countries adopted for the analysis should evidence possibly the biggest similarity in terms of the GDP per capita measure and create within their groups the so-called "convergence clubs" (see f.e. [45]). Data in Table 1 allow us to restrict the analysis to 11 countries. The first group includes: Germany, France, Italy, Spain and United Kingdom (prior to Brexit), and the second group includes: the Netherlands, Belgium, Sweden, Austria, Denmark, Finland. Some countries were excluded, for example: Greece, Portugal, due to a relatively low GDP per capita level and, on the other hand, countries with evidently higher and deviating GDP per capita, such as Ireland and Luxembourg. It should be added that except for the GDP per capita qualification criterion for large and smaller countries, the population criterion was also used, as well as, especially, the job market and not the territory. The 2018 GDP per capita in PPS (EUR) in large countries selected for analysis was in the interval of 28,110 (Spain) to 37,760 (Germany), and in smaller countries—34,230 (Finland) to 39,670 (Denmark). As indicated above, among smaller countries that met the criterion of population, some countries were excluded from the analysis, that is Greece and Portugal with 2018 GDP per capita in PPS of, respectively, EUR 21,050 and EUR 23,810, and on the other hand, Ireland and Luxembourg, with 2018 GDP per capita in PPS of, respectively, EUR 58,650 and EUR 80,870.

Table 1. Selected social economic indicators. Source: own study based on data from: [17].

| Country | Population 2018 (Millions) | Total Employment 15–64 Years, 2018 (per Thousand Persons) | GDP per Capita PPS, 2018 (EUR) | GDP per Capita/PPS Index (EU15, 2008 = 100) | General Government Gross Debt (in % of GDP) |
|---|---|---|---|---|---|
| Germany | 83 | 40.636 | 37.760 | 124 | 61.9 |
| France | 67.0 | 26.744 | 32.070 | 116 | 98.4 |
| Italy | 60.3 | 22.586 | 29.670 | 106 | 134.8 |
| Spain | 46.9 | 19.136 | 28.110 | 107 | 97.6 |
| Netherlands | 17.3 | 8.543 | 39.920 | 108 | 52.4 |
| Belgium | 11.6 | 4.699 | 36.250 | 121 | 100 |
| Sweden | 10.2 | 4.910 | 37.310 | 112 | 38.8 |
| Austria | 8.9 | 4.241 | 39.450 | 120 | 74.0 |
| Denmark | 5.8 | 2.739 | 39.670 | 121 | 34.2 |
| Finland | 5.5 | 2.465 | 34.230 | 108 | 59.0 |
| United Kingdom | 66.6 | 31.112 | 32.570 | 112 | 85.9 |

The referred statistics confirm that in the adopted group of countries, the medium–high level of GDP per capita is evidenced for smaller countries, which in the subject literature are focused on less extensively than large economies, which are commonly considered to be stronger due to the effect of scale. Moreover, it is easy to notice that smaller countries from the selected group are more similar to one another than to the large countries, among which the distance to the leader is greater.

Taking into account the size of the GDP per capita in 2018 in a group of large countries, the difference between the highest and the lowest value was EUR 9650. It is worth noting that in the group of these four countries, as many as three are the founding countries of the current European Union and only Spain (1986) is an example of a country classically catching up. In the group of small countries, however, as many as four, apart from the Netherlands and Belgium (1957) joined the European Union much later (Denmark 1973, the remaining countries 1995), and the difference in GDP per capita between them in 2018 was only EUR 5690. It follows that community policies were not decisive for economic growth alone, and that national governance played a greater role. The transition from economic growth to sustainable growth alone shows mixed performance in both groups of countries, albeit with a better effect on smaller countries in creating sustainable development. The ranking created may be a suggestion for identifying good practices in individual countries.

In the context of the transition from economic growth to sustainable growth, there are quite skeptical conclusions in the literature on the formulation of general causal relationships in this respect. Rather, they concern statements about the very complex relationship between growth and sustainable growth objectives [46] as well as focusing more on homogeneous countries with similar characteristics [47].

Therefore, the idea of analyzing the effectiveness of investments in the implementation of sustainable development indicators adopted in this study seems to be a step towards identifying the differences between countries with the assumption of identifying convergence clubs, linked to both the convergence studies and the size of economies.

On the other hand, the literature points to the use of holistic approaches and GST-based methods of system thinking, which can help achieve better results in the analysis of socio-economic systems. The Grey Systems Theory and related methods can be a very attractive, practical and appropriate way of solving social and economic problems [25].

The convergence clubs adopted for analysis are graphically represented in Figure 1.

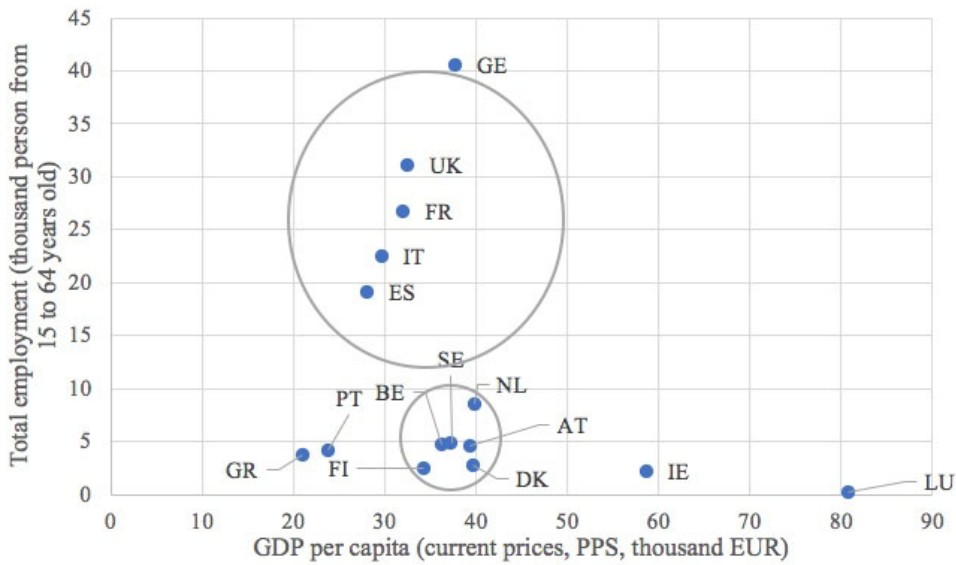

**Figure 1.** Convergence clubs (2018). Source: own calculation based on [17]. Key: GE—Germany, UK—United Kingdom, FR—France, IT—Italy, ES—Spain, GR—Greece, PT—Portugal, BE—Belgium, FI—Finland, SE—Sweden, NL—the Netherlands, AT—Austria, DK—Denmark, IE—Ireland, LU—Luxembourg.

Moreover, it is easy to notice that smaller countries from the selected group are more similar to one another than they are to large countries, in which the distance to the leader is greater, and which clearly evidence a hierarchical position stemming from the convergence process. Following the trial selection, in the steps 1–7, results have been presented for the determination procedure of synthetic indicator for sustainable economic growth.

### 3.1. Step 1. Determination of Basic Elements of Whitenization Model

The countries that were the subjects of analysis were assigned with the following denotations: Germany—$o_1$, United Kingdom—$o_2$, France—$o_3$, Italy—$o_4$, Spain—$o_5$, Sweden—$o_6$, Austria—$o_7$, Denmark—$o_8$, Finland—$o_9$, the Netherlands—$o_{10}$ and Belgium—$o_{11}$. The expenditures in the efficiency model were assigned with the following denotations: GDP per capita in PPS—$i_1$, general government gross debt (percentage of gross domestic product)—$i_2$. The results, however, were assigned with the following denotations: gross domestic expenditure on R&D by sector (% of GDP)—$r_1$, employment in high- and medium-high technology manufacturing and knowledge—$r_2$, R&D personnel by

sector—$r_3$, patent applications to the European Patent Office—$r_4$, share of buses and trains in total passenger transport—$r_5$, share of rail and inland waterways in total freight transport—$r_6$, average $CO_2$ emissions per km from new passenger cars—$r_7$. The research covered calculation of efficiency for all countries in 2016–2018. Results presented in steps 1–7 mirror the 2018 data. If for any of the expenditures or results in the moment of drafting this elaboration the 2018 data were not available, then, according to a naive forecasting model, the 2018 data were adopted as being the same as the 2017 data. Such a situation existed for three results: patent applications to the European Patent Office ($r_4$), share of buses and trains in total passenger transport ($r_5$) and share of rail and inland waterways in total freight transport ($r_6$). In Table 2, input data for the efficiency model pertaining to 2018 were presented.

**Table 2.** Data matrix $D$ for efficiency model (2018). Source: own calculations.

|  | $o_1$ (GER) | $o_2$ (UK) | $o_3$ (FRA) | $o_4$ (ITA) | $o_5$ (ESP) | $o_6$ (SWE) | $o_7$ (AUT) | $o_8$ (DEN) | $o_9$ (FIN) | $o_{10}$ (NED) | $o_{11}$ (BEL) |
|---|---|---|---|---|---|---|---|---|---|---|---|
| $r_1$ | 3.13 | 1.70 | 2.20 | 1.39 | 1.24 | 3.32 | 3.17 | 3.03 | 2.75 | 2.16 | 2.76 |
| $r_2$ | 50.6 | 53.3 | 50.4 | 40.7 | 39.9 | 58.2 | 44.9 | 53.4 | 50.6 | 48.6 | 53 |
| $r_3$ | 1.68 | 1.45 | 1.53 | 1.23 | 0.99 | 1.75 | 1.83 | 2.23 | 1.88 | 1.77 | 1.78 |
| $r_4$ | 228.81 | 82.62 | 141.85 | 68.46 | 35.56 | 283.46 | 231.35 | 246.61 | 235.68 | 203.59 | 145.83 |
| $r_5$ | 14.4 | 13.9 | 17.2 | 18 | 14.8 | 16.7 | 22.3 | 18.5 | 15.8 | 14.3 | 18 |
| $r_6$ | 27.2 | 9.5 | 12.2 | 13.2 | 5 | 31.1 | 33.7 | 11.8 | 29.3 | 49.6 | 27.9 |
| $r_7$ | 129.5 | 124.7 | 112.1 | 115.6 | 118.1 | 122.2 | 123.1 | 109.5 | 116.6 | 105.5 | 119.4 |
| $i_1$ | 37,760 | 32,570 | 32,070 | 29,670 | 28,110 | 37,310 | 39,450 | 39,670 | 34,230 | 39,920 | 36,250 |
| $i_2$ | 61.9 | 85.9 | 98.4 | 134.8 | 97.6 | 38.8 | 74 | 34.2 | 59 | 52.4 | 100 |

### 3.2. Step 2. Elaboration of Scaled Matrices of Input Data $D^*$

Scaling of input data in the elaborated model is necessary for one variable is a minimand, and the rest are maximands. For the purpose of scaling data, Formulas (3) and (4) were used (Table 3).

**Table 3.** Scaled input data matrix $D^*$ (2018). Source: own calculations.

|  | $o_1$ (GER) | $o_2$ (UK) | $o_3$ (FRA) | $o_4$ (ITA) | $o_5$ (ESP) | $o_6$ (SWE) | $o_7$ (AUT) | $o_8$ (DEN) | $o_9$ (FIN) | $o_{10}$ (NED) | $o_{11}$ (BEL) |
|---|---|---|---|---|---|---|---|---|---|---|---|
| $r_1$ | 0.943 | 0.512 | 0.663 | 0.419 | 0.373 | 1.000 | 0.955 | 0.913 | 0.828 | 0.651 | 0.831 |
| $r_2$ | 0.869 | 0.916 | 0.866 | 0.699 | 0.686 | 1.000 | 0.771 | 0.918 | 0.869 | 0.835 | 0.911 |
| $r_3$ | 0.753 | 0.648 | 0.687 | 0.551 | 0.447 | 0.785 | 0.818 | 1.000 | 0.840 | 0.793 | 0.795 |
| $r_4$ | 0.807 | 0.291 | 0.500 | 0.242 | 0.125 | 1.000 | 0.816 | 0.870 | 0.831 | 0.718 | 0.514 |
| $r_5$ | 0.646 | 0.623 | 0.771 | 0.807 | 0.664 | 0.749 | 1.000 | 0.830 | 0.709 | 0.641 | 0.807 |
| $r_6$ | 0.548 | 0.192 | 0.246 | 0.266 | 0.101 | 0.627 | 0.679 | 0.238 | 0.591 | 1.000 | 0.563 |
| $r_7$ | 1.057 | 1.226 | 1.245 | 1.345 | 1.420 | 1.070 | 1.012 | 1.006 | 1.166 | 1.000 | 1.101 |
| $i_1$ | 0.946 | 0.816 | 0.803 | 0.743 | 0.704 | 0.935 | 0.988 | 0.994 | 0.857 | 1.000 | 0.908 |
| $i_2$ | 0.459 | 0.637 | 0.730 | 1.000 | 0.724 | 0.288 | 0.549 | 0.254 | 0.438 | 0.389 | 0.742 |

### 3.3. Step 3. Elaboration of Vector of Synthetic Expenditure Indicators $I_k$ for All Decision-Making Objects

Vector of synthetic expenditure indicators $I_k$ for all decision-making objects is determined with Formulas (6) and (7), and has the following form:

$$I_k = [1.405, 1.453, 1.533, 1.743, 1.428, 1.222, 1.537, 1.247, 1.295, 1.389, 1.650]$$

### 3.4. Step 4. Elaboration of Partial Efficiencies Matrix E

The partial efficiencies matrix E for the analyzed decision-making objects is presented in Table 4.

**Table 4.** Partial efficiencies matrix *E* (2018). Source: own calculations.

|  | $o_1$ (GER) | $o_2$ (UK) | $o_3$ (FRA) | $o_4$ (ITA) | $o_5$ (ESP) | $o_6$ (SWE) | $o_7$ (AUT) | $o_8$ (DEN) | $o_9$ (FIN) | $o_{10}$ (NED) | $o_{11}$ (BEL) |
|---|---|---|---|---|---|---|---|---|---|---|---|
| $e_1$ | 0.75 | 0.19 | 0.33 | 0.00 | 0.04 | 1.00 | 0.66 | 0.85 | 0.69 | 0.40 | 0.46 |
| $e_2$ | 0.52 | 0.55 | 0.39 | 0.00 | 0.19 | 1.00 | 0.24 | 0.80 | 0.65 | 0.48 | 0.36 |
| $e_3$ | 0.46 | 0.27 | 0.28 | 0.01 | 0.00 | 0.67 | 0.45 | 1.00 | 0.69 | 0.53 | 0.35 |
| $e_4$ | 0.67 | 0.15 | 0.33 | 0.07 | 0.00 | 1.00 | 0.61 | 0.83 | 0.76 | 0.59 | 0.31 |
| $e_5$ | 0.13 | 0.00 | 0.31 | 0.14 | 0.15 | 0.78 | 0.94 | 1.00 | 0.50 | 0.14 | 0.26 |
| $e_6$ | 0.49 | 0.09 | 0.14 | 0.13 | 0.00 | 0.68 | 0.57 | 0.18 | 0.59 | 1.00 | 0.42 |
| $e_7$ | 0.28 | 0.55 | 0.46 | 0.34 | 1.00 | 0.65 | 0.00 | 0.44 | 0.72 | 0.18 | 0.03 |

*3.5. Step 5. Determination of Reference and Anti-Reference Vector of Partial Efficiency*

With the use of Formulas (10) and (11), the empirical reference vector (REF) and empirical anti-reference vector (AREF) have been determined.

$$REF = [0.818, 0.818, 0.802, 0.818, 0.665, 0.720, 0.994]$$
$$AREF = [0.240, 0.401, 0.313, 0.088, 0.429, 0.071, 0.658]$$

*3.6. Step 6. Standardization of Partial Efficiencies Matrix E to Interval (0,1)*

The next step of the proposed method consisted in determination of the standardized partial efficiencies matrix $E^*$ (Table 5).

**Table 5.** Standardized partial efficiencies matrix $E^*$ (2018). Source: own calculations.

|  | $o_1$ (GER) | $o_2$ (UK) | $o_3$ (FRA) | $o_4$ (ITA) | $o_5$ (ESP) | $o_6$ (SWE) | $o_7$ (AUT) | $o_8$ (DEN) | $o_9$ (FIN) | $o_{10}$ (NED) | $o_{11}$ (BEL) |
|---|---|---|---|---|---|---|---|---|---|---|---|
| $e_1^*$ | 0.746 | 0.194 | 0.332 | 0.000 | 0.037 | 1.000 | 0.659 | 0.850 | 0.691 | 0.395 | 0.456 |
| $e_2^*$ | 0.522 | 0.550 | 0.392 | 0.000 | 0.189 | 1.000 | 0.242 | 0.802 | 0.648 | 0.480 | 0.362 |
| $e_3^*$ | 0.456 | 0.272 | 0.276 | 0.006 | 0.000 | 0.673 | 0.449 | 1.000 | 0.687 | 0.528 | 0.345 |
| $e_4^*$ | 0.666 | 0.154 | 0.327 | 0.069 | 0.000 | 1.000 | 0.607 | 0.835 | 0.759 | 0.588 | 0.307 |
| $e_5^*$ | 0.130 | 0.000 | 0.314 | 0.144 | 0.151 | 0.778 | 0.939 | 1.000 | 0.500 | 0.139 | 0.255 |
| $e_6^*$ | 0.492 | 0.094 | 0.138 | 0.126 | 0.000 | 0.681 | 0.572 | 0.185 | 0.594 | 1.000 | 0.416 |
| $e_7^*$ | 0.280 | 0.551 | 0.457 | 0.338 | 1.000 | 0.646 | 0.000 | 0.442 | 0.721 | 0.184 | 0.027 |

The result of partial efficiencies matrix E standardization will be the reference vector consisting of 1.000 values only, and the anti-reference vector will consist solely of 0.000 values.

*3.7. Step 7. Determination of Synthetic Indicator for Sustainable Economic Growth for Each Decision-Making Object*

In Table 6, values of synthetic indicator for sustainable economic growth for all decision-making objects have been presented.

**Table 6.** Synthetic indicators for sustainable economic growth for all decision-making objects. Source: own calculations.

|  | $o_1$ (GER) | $o_2$ (UK) | $o_3$ (FRA) | $o_4$ (ITA) | $o_5$ (ESP) | $o_6$ (SWE) | $o_7$ (AUT) | $o_8$ (DEN) | $o_9$ (FIN) | $o_{10}$ (NED) | $o_{11}$ (BEL) |
|---|---|---|---|---|---|---|---|---|---|---|---|
| $E_k$ | 0.204 | 0.065 | 0.099 | 0.010 | 0.006 | 0.677 | 0.235 | 0.542 | 0.431 | 0.176 | 0.086 |

In Figure 2, synthetic indicators for sustainable economic growth for all analyzed countries have been presented in graphical form.

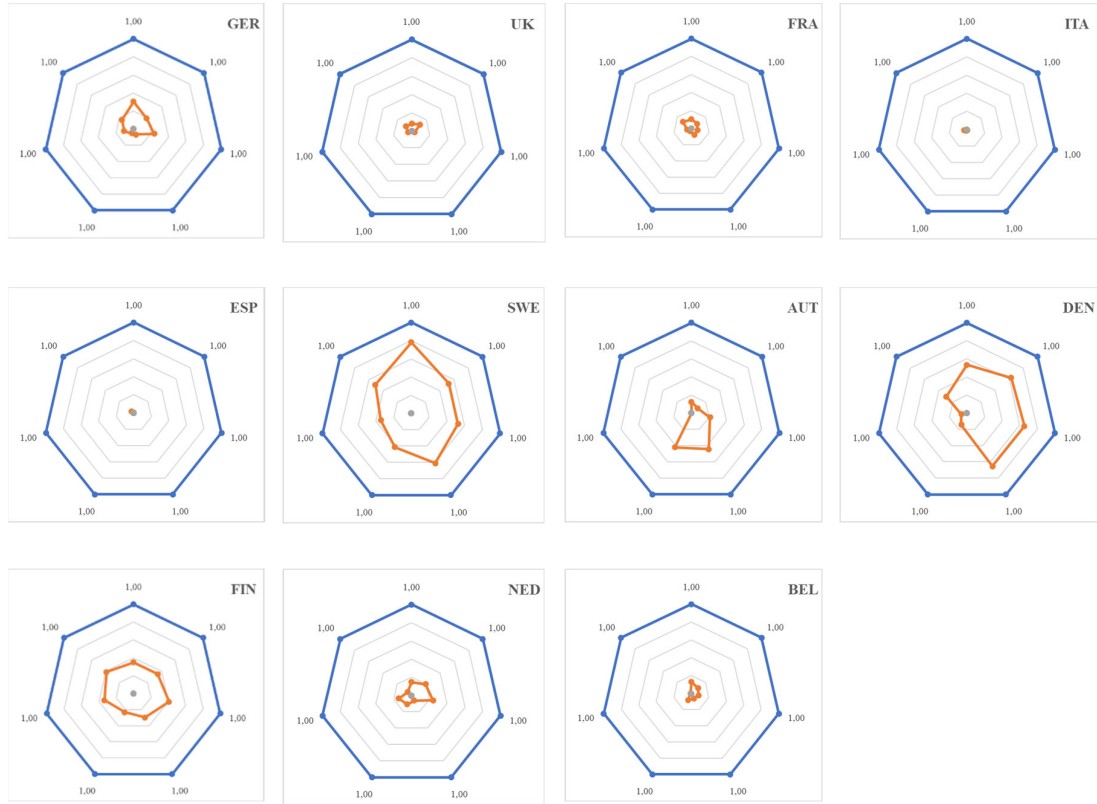

**Figure 2.** Graphical presentation of synthetic indicators for sustainable economic growth for all the analyzed countries in 2018. Source: own study and calculations.

As shown in Figure 2, the area indicated with the position of individual indicators in selected countries is largely differentiated. This pertains both to the area size, as well as its farthest points. There are two leaders—Sweden and Denmark, which are followed by Finland. In Sweden, the highest efficiency is shown by the indicator connected with the gross domestic expenditure on R&D by sector result, and in Denmark, the one connected with the R&D personnel by sector and patent applications to the European Patent Office is the highest. Finland, on the other hand, is characterized by similar values of individual indicators. A considerably lower efficiency of economic growth transformation into sustainable growth is evidenced by large countries, among which relatively better results are shown by Germany alone.

In Table 7, values of synthetic indicators for sustainable economic growth for all decision-making objects have been presented for 2016–2018.

**Table 7.** Synthetic indicators for sustainable economic growth for all decision-making objects (2016–2018). Source: own calculations.

| | $o_1$ | $o_2$ | $o_3$ | $o_4$ | $o_5$ | $o_6$ | $o_7$ | $o_8$ | $o_9$ | $o_{10}$ | $o_{11}$ |
| | (GER) | (UK) | (FRA) | (ITA) | (ESP) | (SWE) | (AUT) | (DEN) | (FIN) | (NED) | (BEL) |
|---|---|---|---|---|---|---|---|---|---|---|---|
| $E_{k_{2018}}$ | 0.204 | 0.065 | 0.099 | 0.010 | 0.006 | 0.677 | 0.235 | 0.542 | 0.431 | 0.176 | 0.086 |
| $E_{k_{2017}}$ | 0.182 | 0.062 | 0.111 | 0.012 | 0.005 | 0.642 | 0.221 | 0.564 | 0.438 | 0.139 | 0.079 |
| $E_{k_{2016}}$ | 0.176 | 0.059 | 0.114 | 0.015 | 0.004 | 0.631 | 0.198 | 0.581 | 0.479 | 0.138 | 0.077 |

The obtained values of synthetic indicators for sustainable economic growth make it possible to elaborate a ranking list of the researched countries for 2016–2018 (Table 8).

**Table 8.** Synthetic indicators for sustainable economic growth for all decision-making objects. Source: own calculations. Ranking list of the researched countries based on "Synthetic Efficiency Indicator for Economic Growth" (SEI-EG) value (2016–2018). Source: own calculations.

| | **2016** | | | **2017** | | | **2018** | |
|---|---|---|---|---|---|---|---|---|
| 1 | Sweden | 0.631 | 1 | Sweden | 0.642 | 1 | Sweden | 0.677 |
| 2 | Denmark | 0.581 | 2 | Denmark | 0.564 | 2 | Denmark | 0.542 |
| 3 | Finland | 0.479 | 3 | Finland | 0.438 | 3 | Finland | 0.431 |
| 4 | Austria | 0.198 | 4 | Austria | 0.221 | 4 | Austria | 0.235 |
| 5 | Germany | 0.176 | 5 | Germany | 0.182 | 5 | Germany | 0.204 |
| 6 | Netherlands | 0.138 | 6 | Netherlands | 0.139 | 6 | Netherlands | 0.176 |
| 7 | France | 0.114 | 7 | France | 0.111 | 7 | France | 0.099 |
| 8 | Belgium | 0.077 | 8 | Belgium | 0.079 | 8 | Belgium | 0.086 |
| 9 | United Kingdom | 0.059 | 9 | United Kingdom | 0.062 | 9 | United Kingdom | 0.065 |
| 10 | Italy | 0.015 | 10 | Italy | 0.012 | 10 | Italy | 0.010 |
| 11 | Spain | 0.004 | 11 | Spain | 0.005 | 11 | Spain | 0.006 |

The first conclusion that might be drawn from the presented ranking list is that it is characterized by high stability throughout the entire period subject to analysis, in terms of position of individual countries in the ranking list. The shape of the synthetic indicator points to countries, in which it is systematically—although to various extents—growing (Sweden, Austria, Germany, the Netherlands, Belgium, the United Kingdom, Spain) and the rest of countries, in which it deteriorates. This probably stems from a relatively short measurement period for this indicator but also from the time-delayed effects of undertaken actions purposed for the transformation of economy towards sustainable growth. Among countries characterized with the highest values of the synthetic development indicator, each time there were the smaller ones. The highest value for this indicator was calculated for Sweden (2016–2018 average amounted to 0.650). The following positions are occupied by Denmark (0.562 on average) and Finland (0.449 on average). High in the ranking list are also Austria (0.218 on average) and Germany that is the highest-ranked large country (0.187 on average). The lower values of synthetic indicator for sustainable economic growth were characteristic for large countries like: Spain (0.005 on average), Italy (0.012 on average) and the United Kingdom (0.062 on average). Spain and Italy are, among the large countries, earlier in their convergence path, which may suggest that they require longer access path to the sustainable growth model. In this case, high debt rate is not a favorable factor in accessing this model. In contrast, the United Kingdom was characterized by both a higher 2018 GDP per capita, and higher average growth rate in the 2008–2018 decade, which points to other factors limiting transition towards the sustainable growth direction. However, with the ceteris paribus assumption, the elaborated ranking list based on the adopted parameters indicates that, in general and in comparison to large countries, smaller countries are characterized by considerably higher transition efficiency of economies towards the sustainable growth direction. Thus, it is legitimate to state that—contrary to large countries, whose economic position, to a large extent, stems from the achieved effect of scale—the smaller countries build their long-term economic advantage more on the basis of sustainable growth. Therefore, the conclusion popular in the latest subject literature about small, but smart countries appears to be confirmed (f.e. [48]).

## 4. Conclusions

The paper presents the concept of a new indicator, exemplifying the conversion efficiency of expenditures in the form of GDP (per capita in PPS) and the general government gross debt (percentage of GDP) into results connected with the sustainable growth in industry, innovation and infrastructure area. The authors dubbed this indicator the "SEI-EG". In the elaboration, a formal (mathematical) model was presented, leading to determination of the synthetic SEI-EG value, based on whitenization functions, i.e., the basic theoretical concept belonging to Grey System Theory. Using the concept of the elaborated indicator, 11 EU15 member states were analyzed in the context of

efficiency in obtaining results towards sustainable economic growth. In the analyzed group of countries, there were smaller countries—Sweden, Denmark, Finland, Austria, the Netherlands and Belgium, as well as large countries—Germany, the United Kingdom, France, Spain and Italy. The qualification criterion for a country to be classified as large or smaller was not the territory but the job market connected with the generated GDP per capita. Analysis results showed that smaller countries are in the lead of the ranking list of the efficiency determined in this way. Large countries, like Spain, Italy and the United Kingdom are, conversely, at the bottom of the ranking list. A sort of exception in this respect is Germany, for which the efficiency in the analyzed period was higher than in some smaller countries, like Belgium and the Netherlands. However, Germany was not in the top three, no matter the year; the top three positions were dominated by the Nordic countries — Sweden, Denmark and Finland. This draws attention to these countries, for they may become a sort of benchmark for economic policy in other countries. The newly elaborated indicator may be applied in practice for the purpose of efficiency analysis of economic expenditure transformation into results pertaining to sustainable economic growth for many country groups. From an economic policy point of view, attention should be paid to the relatively best results achieved in individual countries under the various indicators and attempts should be made to exchange experiences in the framework of good practice, since sustainable development must be based on political will. Organized institutional cooperation is therefore needed. The institutional element is considered to be the most important to achieve exchange between economic and environmental issues. "The overarching aim is to meet wider economic and social needs, while limiting environmental impact and realizing reductions in harmful emissions. The institutional component has been recognized as the most important for achieving trade-of between economic issues, and environmental ones" [49]. At the same time, the indicator features limitation connected with the fact that it is a relative measure, and its value is the function of the as-is empirical base. If the group of the analyzed countries is to be broadened with subsequent sets, there is the need to re-calculate the entire calculation model, which stems from the changed empirical base. The model may also be adapted to an altered set of expenditures and results. This circumstance may be the subject of subsequent research.

**Author Contributions:** Conceptualization, M.K. and M.N.; data curation, P.Ł. and M.N.; formal analysis, M.N.; resources, P.Ł. and M.K.; validation, M.N.; writing—original draft, P.Ł., M.K. and M.N.; writing—review and editing, P.Ł., M.K. and M.N. All authors have read and agreed to the published version of the manuscript.

**Funding:** This research received no external funding.

**Conflicts of Interest:** The authors declare no conflict of interest.

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
