# Peer review of "Measuring the Efficiency of Economic Growth towards Sustainable Growth with Grey System Theory"

_sustainability, doi:10.3390/su122310121_

Round 1
Reviewer 1 Report
This paper seems to have meet its objectives in ascertaining the facting determining efficeincy towards economic growth and to that effect sustainable around some selected EU countries. Its only notoable deficiency is its unability to accommodate conclusivity when countries with extremes lower or higher GDPs as you mentioned in lines 248 to 250 of your manuscript. However, this is okay as most models do have low tolerrance to outlier values, which deviate from when is deemed normal as compared to other values.
Overally, your paper is good and unique, and is perves ways for more research in related fields and I would recommend it for processing to this next round.
Author Response
The first reviewer evaluated the article unambiguously positively, not indicating the need to introduce any further revisions.
Reviewer 2 Report
The paper Measuring the Efficiency of Economic Growth Towards Sustainable Growth with Grey System Theory wants to provide an indicator for efficiency of expenditures towards economic growth into results pertaining to sustainable development.
I have major concerns about Introduction and Results in terms of implications. I have some comments accordingly:
- The Authors just slightly mention the problem of convergence. However, the problem of convergence (Phillips and Sul, 2007), within-continent and Country differences (Cartone et al. 2020), and Cohesion cannot be discarded (Barca et al. 2020). The economic part should be integrated in the Introduction.
- The methodological soundness go Grey System Theories in general is crystal clear. But what is the value added for the specific research question? This should emerges from Introduction.
- What is the relevance of this study in terms of sustainable development? the Authors should do big efforts to improve implications about sustainability. The conclusion that small countries are more manageable in this direction it is not very original. An example could be in (Coscieme et al., 2020) and included references.
- More attentions should be given by potential assumptions linked by the applications of Grey Systems theories at a minimum.
Therefore, I would suggest major revisions before acceptance.
References
Barca, F., McCann, P., & Rodríguez‐Pose, A. (2012). The case for regional development intervention: place‐based versus place‐neutral approaches. Journal of regional science, 52(1), 134-152.
Cartone, A., Postiglione, P., & Hewings, G. J. (2020). Does economic convergence hold? A spatial quantile analysis on European regions. Economic Modelling.
Coscieme, L., Mortensen, L. F., Anderson, S., Ward, J., Donohue, I., & Sutton, P. C. (2020). Going beyond gross domestic product as an indicator to bring coherence to the sustainable development goals. Journal of Cleaner Production, 248, 119232.
Phillips, P. C., & Sul, D. (2007). Transition modeling and econometric convergence tests. Econometrica, 75(6), 1771-1855.
Author Response
- Comment: “The Authors just slightly mention the problem of convergence. However, the problem of convergence (Phillips and Sul, 2007), within-continent and Country differences (Cartone et al. 2020), and Cohesion cannot be discarded (Barca et al. 2020). The economic part should be integrated in the Introduction”.
The Authors significantly expanded the description of the convergence issue and included it in the introduction. Among other things, all references to the literature indicated by the reviewer were included.
- Comment: “The methodological soundness go Grey System Theories in general is crystal clear. But what is the value added for the specific research question? This should emerges from Introduction”.
The Authors indicated in the Introduction the value added for the undertaken research question related to the use of grey systems theory
- Comment: “What is the relevance of this study in terms of sustainable development? the Authors should do big efforts to improve implications about sustainability. The conclusion that small countries are more manageable in this direction it is not very original. An example could be in (Coscieme et al., 2020) and included references”.
The authors demonstrated the importance of the issues addressed in the article in the context of sustainable development. The article was supplemented with all the issues indicated by the Reviewer in the paper (Coscieme et al., 2020) and in other sources.
- Comment: “More attentions should be given by potential assumptions linked by the applications of Grey Systems theories at a minimum.’’
The authors developed a description of the Grey Systems Theory - both in terms of its essence, its application in modelling uncertainty in socio-economics systems, and the added value resulting from the application of GST methodology to the research problem under consideration.
Reviewer 3 Report
Title: Measuring the Efficiency of Economic Growth Towards Sustainable Growth with Grey System Theory
Comments to the Authors
General Comment
The paper proposes a Synthetic Efficiency Indicator for Economic Growth to measure outcomes of economic growth with benefits toward sustainable economic growth. The proposed indicator is defined based on withenization functions, basic concepts in Grey System Theory, and is used to elaborate a ranking list of 15 EU countries based on the efficiency of economic growth towards sustainable growth.
The paper deals with an interesting topic but disregards some previous literature on the application of Grey System Theory in the analysis of socioeconomic systems. Some methodological choices are not fully justified and results from the empirical analysis are not clearly explained. Policies implications of the obtained results are not properly highlighted. Overall, the limited novel contribution and a number of methodological concerns prevent me to recommend the publication of the manuscript in the current shape. Major revisions are required.
Specific Comments
Abstract
Abstract should be rewritten to better emphasize the novel contribution of the paper and to highlight the motivation of the proposed analysis.
Keywords
Keywords should not overlap with the title.
Introduction
- The definition of sustainable development and its link with economic growth should be better clarified. Some previous literature on this topic should be considered (see, e.g., Islam et al. 2003).
- The notion of sustainable development and the choice of indicators should be better linked to the EU sustainable development goals (see, European Commission, 2010; European Union, 2019).
- The contribution of the manuscript to both the theoretical and the empirical analysis should be highlighted.
Material and Methods
Some previous literature on the application of the Grey System Theory in the analysis of socioeconomic systems should be considered (for an overview, see, e.g., Javanmardi and Liu, 2019).
Results
- The use of the distinction between smaller and larger countries for the identification of two clusters of EU countries should be better motivated and linked to a better identified research question. Alternative criteria for the identification of clusters could be also taken into consideration.
- The focus on 15 EU countries instead of 28 EU countries should be better motivated. The opportunity of considering the regional level of analysis should be also discussed (see, e.g., Malik and Ciesielska, 2011; Mirshojaeian Hosseini and Kaneko, 2012; Jovovic et al. 2017).
- Differences and similarities with empirical findings in some previous literature should be emphasized.
- More emphasis should be given to the policy implications of the empirical findings. On the relationship between country level governance, economic growth and sustainable development see, e.g., Boţa-Avram et al. (2018).
Conclusions
Interpretation of results and policy suggestions in the conclusions are not that informative.
Other comments
The manuscript contains some typos:
- On line 246 (page 7) Tabele should be replaced by Table.
- On line 292 (page 9) the dot after Table 2 in the text should be removed.
- On line 325 (page 11) the dot after Figure 2 in the text should be removed.
- On line 338 (page 11) the dot after Table 7 in the text should be removed.
References
Boţa-Avram, C., Groşanu, A., Răchişan, P. R., & Gavriletea, M. (2018). The bidirectional causality between country-level governance, economic growth and sustainable development: A cross-country data analysis. Sustainability, 10(2), 502. https://doi.org/10. 3390/su10020502.
European Commission (2010). Europe 2020 – A strategy for smart, sustainable and inclusive growth. Brussels.
European Union (2019). Smarter, greener, more inclusive? Indicators to support the Europe 2020 strategy - 2019 edition. Luxembourg, Publication Office of the European Union.
Islam, S.M.N., Munasinghe, M., & Clarke, M. (2003). Making long term economic growth more sustainable: evaluating the costs and benefits. Ecological Economics, 47 (2-3), 149-166.
Javanmardi, E. & Liu, S. (2019). Exploring grey system theory-based methods and applications in analyzing socio-economic systems. Sustainability, 11(15), 4192; doi:10.3390/su11154192.
Jovovic, R., Draskovic, M., Delibasic, M., & Jovovic, M. The concept of sustainable regional development – institutional aspects, policies and prospects. Journal of International Studies, 10(1), 255-266.
Malik, K., & Ciesielska, M. (2011). Sustainability within the region: the role of institutional governance. Economic and Environmental Studies, 11(2), 167-187.
Mirshojaeian Hosseini, H. & Kaneko, S. (2012). Causality between pillars of sustainable development: Global stylized facts or regional phenomena? Ecological Indicators, 14(1), 197-201.
Author Response
Introduction:
- Comment: “The definition of sustainable development and its link with economic growth should be better clarified. Some previous literature on this topic should be considered (see, e.g., Islam et al. 2003)”.
The Authors explained the definition of sustainable development in more detail and identified its links to the issue of economic growth. The scope of the cited literature was significantly expanded. It includes the publications suggested by the Reviewer.
- Comment: “The notion of sustainable development and the choice of indicators should be better linked to the EU sustainable development goals (see, European Commission, 2010; European Union, 2019)”.
The authors demonstrated the link between the indicators selected for the study and the publications on sustainable development published by the European Union institutions.
- Comment: ’’The contribution of the manuscript to both the theoretical and the empirical analysis should be highlighted”.
The authors emphasized both the theoretical and empirical contribution of the article to the development of the field of sustainable development.
Materials and Methods:
- Comment: “Some previous literature on the application of the Grey System Theory in the analysis of socioeconomic systems should be considered (for an overview, see, e.g., Javanmardi and Liu, 2019)”.
The authors developed a description of the Grey Systems Theory - both in terms of its essence, its application in modelling uncertainty in socio-economics systems, and the added value resulting from the application of GST methodology to the research problem under consideration. Reference was made, among others, to the literature position indicated by the Reviewer, in which the problem was presented in detail.
Results:
- Comment: “The use of the distinction between smaller and larger countries for the identification of two clusters of EU countries should be better motivated and linked to a better identified research question. Alternative criteria for the identification of clusters could be also taken into consideration”.
The authors explained the broader division of the countries under examination into two convergence groups. The relationship between the research problem undertaken and the adopted division was emphasized. The choice of decision criteria was justified and it was pointed out that the model under study allows for the application of either additional or different decision criteria.
- Comment: “The focus on 15 EU countries instead of 28 EU countries should be better motivated. The opportunity of considering the regional level of analysis should be also discussed (see, e.g., Malik and Ciesielska, 2011; Mirshojaeian Hosseini and Kaneko, 2012; Jovovic et al. 2017)”.
The authors explained in more detail the criteria for selecting countries in the research sample. The possibility of regional differentiation in the model used was also presented.
- Comment: “Differences and similarities with empirical findings in some previous literature should be emphasized”.
The authors pointed out the similarities and differences between the obtained results and analogous research results presented in the literature.
- Comment: “More emphasis should be given to the policy implications of the empirical findings. On the relationship between country level governance, economic growth and sustainable development see, e.g., Boţa-Avram et al. (2018)”.
The authors presented the policy recommendations resulting from their research in more detail.
Conclusions:
- Comment: „Interpretation of results and policy suggestions in the conclusions are not that informative”.
The authors rewrote the Conclusions section in accordance with the comments.
Round 2
Reviewer 2 Report
All comments have been addressed. Therefor, I would suggest acceptance with minus English check.
Reviewer 3 Report
I would like to thank the authors for having gone through the first round of comments and I appreciate their intents to improve the manuscript. I have no major remarks to this version, so I suggest to accept manuscript in its current form.